# Antidiabetic Wound Dressing Materials Based on Cellulosic Fabrics Loaded with Zinc Oxide Nanoparticles Synthesized by Solid-State Method

**DOI:** 10.3390/polym14112168

**Published:** 2022-05-27

**Authors:** Hany Elsawy, Azza Sedky, Manal F. Abou Taleb, Mohamed H. El-Newehy

**Affiliations:** 1Department of Chemistry, College of Science, King Faisal University, Al-Ahsa 31982, Saudi Arabia; 2Department of Chemistry, Faculty of Science, Tanta University, Tanta 31527, Egypt; 3Department of Biological Sciences, College of Science, King Faisal University, Al-Ahsa 31982, Saudi Arabia; asedky@kfu.edu.sa; 4Department of Zoology, Faculty of Science, Alexandria University, Alexandria 21548, Egypt; 5Department of Chemistry, College of Science and Humanities, Prince Sattam Bin Abdulaziz University, Al-Kharj 11942, Saudi Arabia; abutalib_m@yahoo.com; 6National Center for Radiation Research and Technology (NCRRT), Department of Polymer Chemistry, Egyptian Atomic Energy Authority, Nasr City, Cairo 11762, Egypt; 7Department of Chemistry, College of Science, King Saud University, Riyadh 11451, Saudi Arabia

**Keywords:** cellulosic fabrics, zinc oxide nanoparticles, wound dressing, antidiabetic

## Abstract

The current study aims for the use of the solid-state technique as an efficient way for the preparation of zinc oxide nanoparticles (ZnONPs) as an antimicrobial agent with high concentration using sodium alginate as stabilizing agent. ZnONPs were prepared with three different concentrations: ZnONPs-1, ZnONPs-2, and ZnONPs-3 (attributed to the utilized different concentrations of zinc acetate, 1.5, 3 and 4.5 g, respectively). The as-fabricated ZnONPs (ZnONPs-1, ZnONPs-2, and ZnONPs-3) were used for the treatment of cellulosic fabrics as dressing materials for the diabetic wounds. DLS findings illustrated that the as-prepared ZnONPs exhibited average particle size equal to 78, 117, and 144 nm, respectively. The data also showed that all the formulated ZnONPs were formed with good stability (above −30 mv). The topographical images of cellulosic fabrics loaded with ZnONPs that were obtained by SEM confirmed the deposition of nanoparticles onto the surface of cellulosic fabrics with no noticeable agglomeration. The findings also outlined that the treated cellulosic fabrics dressings were proven to have enhanced bactericidal characteristics against the pathogenic microorganisms. The finding of wound contraction for the diabetic rats was measured after 21 days and reached 93.5% after treating the diabetic wound with cotton fabrics containing ZnONPs-2. Ultimately, the generated wound dressing (cellulosic fabrics loaded with ZnONPs) offers considerable promise for treating the wound infections and might be examined as a viable alternative to antibiotics and topical wound treatments.

## 1. Introduction

Textile materials have been utilized and created for medicinal applications since the dawn of civilization [1]. In recent years, textile materials have been significantly included in a variety of uses and more advanced alternatives for today. Meanwhile, antibacterial fabrics are used in various forms, including protective fiber and bed sheets to restrict the spread of illnesses, and their use in medical applications has expanded quickly due to their outstanding qualities [2,3,4]. Moreover, the rapid use of nanomaterials in the textile fabrics and apparel industry has accelerated this massive change [5]. The treatment of textile fabrics with nanomaterials such as nanoparticles (NPs) can improve their antimicrobial properties, which can be used in wound dressings [6]. In addition, the most recent wound dressings are meant to serve many purposes, including not just covering the wound but also assisting in the healing process [5,7,8]. The healing process has multiple parts, and any disruption in any of them might result in serious problems. Bacterial infections are the most common problem that affects wound healing, and they can have serious long-term consequence [9,10].

To really be effective, wound dressings should typically meet three key criteria: maintaining the cleaning of wounds and their absence of infectious diseases, preserving wound moisture to aid healing and reduce maceration, and removing the fluid exudates and toxic components via their absorptions [11]. However, with some wound dressings saturated with human fluids, quick bacterial growth can occur, resulting in serious infections that are threatening to the patient’s health and life [12]. To allow the timely healing of cutaneous wounds, it is critical to keep the wound free from pathogenic bacteria [13]. Antibiotics used in conjunction with enzymes, metal ions, and quaternary ammonium compounds can result in antibiotic resistance, environmental harm, and expensive costs. As a result, new bactericidal activity approaches have been developed. Therefore, the utilization of cotton fabrics loaded with nanoparticles as wound dressings is becoming more common.

The utilization of nanoparticles for textile finishing is a fantastic idea because of their great efficiency in focused action, which improves the efficacy of materials [14,15]. Zinc oxide nanocrystals (ZnONPs) are multifunctional nanoparticles that have attracted interest owing to their unique combination of electrical, physical, biological, chemical, optical, and long-term environmental stability, biocompatibility, cheap cost, and non-toxic potentials [16,17,18].

Regarding the in situ fabrication, Sun et al. [19] produced bactericidal ZnONPs-loaded cotton fabric using ultrasonic irradiation in which ZnONPs were synthesized with the aid of amino-terminated silicon solution. Such studies have described a technique for the deposition of nanostructured ZnONPs onto cotton fabric to introduce advanced antibacterial features. Their findings revealed that ZnONPs-loaded cotton fabric had extremely good UV protection and remarkable antibacterial characteristics. Recently, Sun X et al. [20] established an in-situ synthesis technique for ZnONPs to be used for the producing of bactericidal fabrics for healing domains. Their findings revealed that the fabric containing ZnONPs had absolutely superb UV protection, efficient antimicrobial functions, as well as modest cytotoxicity. Lumbreras-Aguayo et al. [21] create bactericidal cotton gauze fabrics, which are chemically modified through graft copolymerization with poly (methacrylic acid). After that, ZnONPs were added to the grafted cotton gauzes to give them bactericidal features.

The current scientific effort demonstrated an effective, high yield, low cost, and environmentally acceptable method for solid-state fabrication of ZnONPs without the need for organic or inorganic solvents [22] using sodium alginate, benign carbohydrate polymer, as stabilizing agent. Antimicrobial properties of ZnONPs-loaded cellulosic cotton fabrics were assessed against four microbes: *E. coli* and *S. aureus*, *C. albicans*, and *A. niger*. Ultimately, the treated wounds’ histological characteristics were assessed.

## 2. Materials and Methods

### 2.1. Materials

Zinc acetate (Zn (CH_3_COO)_2_) as a precursor for zinc oxide nanoparticles was purchased from Sigma Aldrich Co, Dresden, Germany. Meanwhile, sodium hydroxide, and sodium alginate were purchased from Across Co., St. Louis, MO, USA. Streptozotocin (STZ) and homocysteine standard were purchased from Sigma Chemical Co., St. Louis, MO, USA. Deionized water was used for dilution, analysis, and application. All other chemicals are of analytical grade and were used as received without further purification or modification.

### 2.2. Animals

A total of forty male Sprague-Dawley rats weighing 170 g were utilized to evaluate the healing efficiency of ZnONPs-treated cellulosic cotton fabrics. For seven days, all rats were kept individually in sterile stainless-steel cages under normal temperature and light standards and fed standard rodent chow. The procedure for animal treatments were reviewed and approved by the research ethics committee at King Faisal University (Ref. No. KFU-REC-2022-FEB-EA000436/2022-02-15).

### 2.3. Methods

#### 2.3.1. Preparation of Solid-State Zinc Oxide Nanoparticles (ZnONPs)

Three concentrations of ZnONPs were prepared as follows: zinc acetate (1.5 g) was added to sodium hydroxide (0.9 g) and mechanical grinding was continued, and then sodium alginate (0.9 g) was added and mechanically blended and grinded for 5 min at ambient temperature. Two more fine homogeneous solid mixtures were acquired by repeating this step with another two concentrations of zinc acetate (3 g and 4.5 g) (Figure 1).

The three fine powders were subjected to washing with distilled water and submitted to a centrifugation step to acquire pure powders of ZnONPs stabilized or coated with sodium alginate. Finally, the wet, fine powders of ZnONPs were dried using a freeze-drying instrument. The three different concentrations were coded, depending on the concentration of zinc acetate (1.5, 3 and 4.5 g), as ZnONPs-1, ZnONPs-2, and ZnONPs-3, respectively.

#### 2.3.2. Treatment of Cellulosic Fabrics with the As-Prepared ZnONPs

A known weight (1 g) of each sample (ZnONPs-1, ZnONPs-2, and ZnONPs-3) was dissolved in deionized water (100 mL) under the effect of sonication for 15 min. The bleached cotton fabric (10 × 10 cm) was dipped in each solution for 1 min and squeezed with pick up 80%. The treated cotton fabrics were dried at 90 °C for 7 min and cured at 150 °C for 3 min.

#### 2.3.3. Induction of Diabetes and Incision Wound Operation

Streptozotocin (STZ) was dispersed in sodium citrate solution (50 mM; pH 4.5) comprising NaCl (150 mM) and injected subcutaneously into rats (3 mg/50 g of body weight). Fasting blood sugar was measured after three days to affirm the establishment of diabetes mellitus. If the animal’s fasting glucose level was higher than 200 mg/dL, they became catalogued as a diabetic animal [23]. Diethyl ether (45%) was used to anaesthetize the diabetic rats. Ethanol (70%) was used to sterilize the wounded area (shaved area). Taking that in mind, the single longitudinal skin incision was performed on just the shaved area.

#### 2.3.4. Design of the Experimental Wounded Rats

The wounded skin rats were evenly dressed with an experimental sterile dressing consisting of cellulosic fabric loaded with ZnONPs in various concentrations: ZnONPs-1@CF (Group II), ZnONPs-2@CF (Group III), and ZnONPs-3@CF (Group IV) per kg b.w. Taking that into mind, the negative control group was kept with untreated cotton materials that were not treated in any way (Group I). This experiment took place over the course of 21 days. Daily, the animals were treated with a new sample of the treated cotton fabric that was appropriate for each group.

### 2.4. Characterization of the ZnONPs, ZnONPs-Treated Cotton Fabrics, and the Diabetic Rats Dressed with ZnONPs-Cotton Fabrics

UV-vis spectra were utilized to define the wavelength and exact absorption of ZnONPs. The generated dispersed solution was evaluated using PG Instrumentals Ltd. (Germany). On the other hand, the shape and distribution of ZnONPs were assessed via TEM (JEM-1200, JEOL, Tokyo, Japan).

Dynamic Light Scattering (DLS) is a one-of-a-kind method for calculating the average hydrodynamic size of ZnONPs based on their intensity distribution (called the Z average). As a result, Nano-ZS, Malvern Instruments Ltd. (UK) was used to calculate the average diameter, size distribution, and zeta potential of samples of ZnONPs. The obtained values of zeta potential indicate the stability of the nanoparticles that have been synthesized. A high zeta potential (more than ±30 mv) confers stability to molecules and particles of sufficient size, i.e., the solution or dispersion will not aggregate. It is an important precaution that the samples are subjected to sonication for 20 min before evaluation. The topographical features of the cellulosic fabric that are native or treated with different concentrations of ZnONPs were assessed via field emission scanning electron microscopy (FESEM; Quanta FEG 250; FEI Ltd., Brno, Czech Republic).

### 2.5. Antimicrobial Properties Cellulosic Fabrics Loaded with ZnONPs

The bactericidal efficiency of untreated and treated cellulosic fabrics loaded with ZnONPs (with different concentrations) were determined using the disc agar plate technique [24,25]. Antimicrobial activities were evaluated against most of the known dangerous species, including *S. aureus*, *E. coli*, *C. albicans*, and *A. niger*.

The progressive changes in wound area (mm) were recorded on the 7th and 21st days of the experimental period and the size of the wound area was photographed. The percentage of wound contraction was also calculated from the equation that was described previously [26].

At the end of the experiment (after 21 days), the wounds were studied histopathologically as follows: animals were euthanized, the skin was enucleated and soaked in formalin solution (10%), implanted in paraffin, and vertically sectioned (3 cm) for histological examination. An optical microscope with a connected digital camera was utilized for obtaining images of the epidermal–dermal junction for all animals at each time point (magnification is 100×).

## 3. Results and Discussions

The present work has two main purposes focusing on the environmental issues during the preparation and application. Firstly, ZnONPs were synthesized in the solid-state form with high concentrations using sodium alginate as a capping agent. The advantage of preparation using the solid-state technique is mainly focused on the ease of preparing a high concentration, a controlled size with no aggregation, no need for using more than one capping agent, and ease of transportation. The ease of preparation is attributed to the manual blending of all precursors (zinc salt, sodium hydroxide, and sodium alginate), thus there is no need for consuming energy. In addition, by using this method, the as-prepared nanoparticles can be easily transferred to industrial companies or factories that use them in the fabric treatment, unlike nanoparticles in their liquid form, which require large containers in order to be packaged and transported, which increases the cost of production. Furthermore, the use of liquids and solvents in the preparation also increases the cost of preparation. Below are the obtained data for all each step. The grinding of all precursors facilitate the conversion of zin salt to zinc hydroxide coated by sodium alginate. The washing and centrifugation steps help to remove the undesired and excess amount of all precursors that will be dissolved in the distilled water (washing solvent) and easily decanted. Secondly, the as-prepared ZnONPs with different concentrations (depending on the concentration of the utilized zinc salt) were used for the treatment of the cellulosic fabrics. Finally, these treated cellulosic fabrics were used as dressing materials for the diabetic wounds.

### 3.1. Characterization of the As-Synthesized ZnONPs

#### 3.1.1. UV-Vis Spectroscopy of ZnONPs

The wavelength and absorbance of the ZnONPs was measured using UV-Vis spectroscopy. As is widely known, ZnONPs exhibit an absorption band in the range of 200 to 500 nm. The UV-vis spectroscopy of sodium alginate and ZnONPs coated with sodium alginate is shown in Figure 2. UV-vis spectroscopy of ZnONPs revealed a featureless surface plasmon resonance band (SPR) around 360 nm, demonstrating that ZnONPs are formed.

The sample coded with ZnONPs-1 has an absorption peak at 370 nm, while the graph shows that the absorption peak of ZnONPs-2 was moved to 358 nm with increasing the concentration of zinc acetate to 3 g. On the other hand, the absorption peak of ZnONPs-3 was shifted again to a high wavelength (360 nm) when the zinc acetate concentration was increased to 4.5 g. The band is practically sharp with the second concentration (ZnONPs-2), implying that the size of particles seems to be nearly identical and hence the particle generated with homogeneity characteristics. On the other hand, ZnONPs-1 and ZnONPs-3 have very broad bands, indicating that the produced particles are heterogeneous.

#### 3.1.2. TEM of ZnONPs

The powder of the created ZnONPs (ZnONPs-1, ZnONPs-2, ZnONPs-3) (0.05 g) was dispersed in deionized water for particle shape investigation, and the resultant images are shown in Figure 3. For ZnONPs-1 and ZnONPs-2, the particles have a spherical shape and an excellent distribution, indicating that 0.9 g of sodium alginate is suitable for stabilization, and thus leads to the formation of nanoparticles of small size and good stability (Figure 3A,B). On the other hand, ZnONPs-3 exhibited agglomerated particles with no discernible edges because the availability of functional groups in sodium alginate is insufficient to stabilize the nanoparticles of ZnO and unable to protect them completely from aggregation (Figure 3C). The interplanar distance of the fringes of ZnONPs-2 as illustrated from high resolution TEM is 0.28 nm (Figure 3D). The clear fringe finding implies that the nanoparticles are presented as single crystalline structures. Based on the TEM observation, the ZnONPs-2 were chosen for further investigation to examine the generated particles’ selected region diffraction, which revealed that the synthesized ZnONPs-2 were developed as single crystals.

#### 3.1.3. DLS of ZnONPs

DLS was employed to estimate the average mean size of the three created ZnONPs with varying concentrations due to the different amounts of the utilized zinc acetate, and the graphs are shown in Figure 4. It is obvious that the particle size average is consistently changed in tandem with the zinc salt concentration used. ZnONPs-1 have an average particle size of 78 nm, whereas ZnONPs-2 have a particle size average of 117 nm. On the other hand, ZnONPs-3 showed that the average diameter is 144 nm. The differences in particle size throughout the three samples can be related to sodium alginate’s potential as a ZnONPs stabilizer.

Because of the utilization of zinc acetate with low concentration (1.5 g), sodium alginate’s ability to protect the produced ZnONPs-1 is enhanced. When the zinc acetate concentration is increased to 3 g (ZnONPs-2), the potential impact of sodium alginate as a stabilizing agent is largely reduced, resulting in a larger formed particle size. The concentration of zinc oxide ions rises considerably whenever the zinc acetate concentration is increased to 4.5 g (ZnONPs-3) compared to the content of sodium alginate; therefore, its ability to stabilize the formed nanoparticles declines, resulting in an increase in size diameter to 144 nm.

#### 3.1.4. Zeta Potential of ZnONPs

The zeta potential value for each sample is determined from the aforementioned observations and characterization. The zeta potential for the prepared ZnONPs was illustrated in Figure 5. The stabilizing effect is thought to be inversely proportional to the diameter of the size. Because the polymer used has a limited stabilizing impact, an increased particle size was observed while using a high concentration of ZnONPs. The stabilizing effect of sodium alginate is reduced in our study due to the greater concentration of the utilized zinc acetate (above 1.5 g). As described earlier, and as displayed in Figure 5, the zeta potential values for ZnONPs-1 and ZnONPs-2 are −37.5 and −33.7 mv, respectively. Meanwhile, the zeta potential of ZnONPs-3 has two distinct values: -48.7 and -31.4 mv, indicating that the produced ZnONPs-3 is heterogeneous. Additionally, it is well established that ZnONPs-2 solution has a better stability than the other two concentrations.

Generally, as the utilized concentration of zinc acetate was increased, the zeta potential value decreased. However, all obtained values for zeta potentials are above -30 mv, which indicates a strong stability for the produced ZnONPs that is related to the sodium alginate stabilization potential. Moreover, the zeta potential values for all three formed nanoparticles (ZnONPs-1, ZnONPs-2 and ZnONPs-3) exhibit good stability even while using a high concentration of zinc acetate, demonstrating that using solid-state technique is a favorable process for the high-throughput fabrication ZnONPs in the presence of stabilizing agent (sodium alginate). Bear in mind that the negative signals are owing to the hydroxyl and carboxylate groups of sodium alginate having a negative effect.

### 3.2. Characterization of Cellulosic Fabrics Treated with Different Concentrations of ZnONPs

The three prepared nanoparticles (ZnONPs-1, ZnONPs-2, and ZnONPs-3) were used to impart treated cellulosic cotton fabrics coded as ZnONPs-1@CF, ZnONPs-2@CF, and ZnONPs-3@CF, respectively. As revealed from the experimental part, the accurate weight of these nanoparticles was dispersed in distilled water and used as a finishing path for fabrics. After that, the treated fabrics were dried and cured at high temperature to facilitate the crosslinking of nanoparticles with OH groups of cellulosic chains. In addition, most of these nanoparticles are expected to be physically adsorbed onto the surface of cellulosic fabrics. To illustrate these phenomena, SEM was conducted to assess the structure of the surface for the treated cotton fabrics. Thus, Figure 6 shows SEM images of ZnONPs-treated cotton fabrics in comparing with the surface of untreated fabric. All samples were scanned using two different magnifications.

As can be clearly observed in Figure 6A, the untreated cotton fabric (CF) exhibits a smooth surface free from deposited particles onto the fabric surface. Meanwhile, as depicted in Figure 6B–D, the morphological surface of cotton fabric was changed to a rough surface due to the deposition of ZnONPs onto the surface of cotton fabrics. The data obtained from SEM are in agreement with that of TEM and DLS data. Using ZnONPs-1 for fabric treatment leads to the deposition of very small particles onto the surface (Figure 6B). The increase in the concentration of the utilized ZnONPs-2 (Figure 6C) causes significant deposition of aggregated nanoparticles onto the surface. On the other hand, the use of a higher concentration of ZnONPs-3 (Figure 6D) leads to the deposition of aggregated particles. Therefore, the roughness and features of the cellulosic surface was changed depending on the deposited amount and size of the finishing agent (ZnONPs).

The as-prepared cellulosic fabrics loaded with ZnONPs-2 were selected to evaluate the durability of the fabric after many washing cycles. Firstly, the treated fabric was washed with distilled water using traditional detergent for 75 min in a washing machine at 40 °C. Next, the washed fabric was squeezed and dried in air, and then the morphological features were assessed using FESEM. Figure 7A,B illustrates the topographical surface of the treated cellulosic fabrics after washing. The scans of the treated fabrics were evaluated at different magnifications: 2500× (Figure 7A) and 10,000× (Figure 7B). It can be observed from the two images that the nanoparticles are still deposited onto the surface of the fabrics, which can be ascribed to the good adhesion of nanoparticles to the surface of cellulosic fabrics. As mentioned, ZnONPs were prepared using sodium alginate as stabilizing agent to cover and coat these particles and prevent them from agglomeration; therefore, the covered layer (sodium alginate) is able to chemically bond with cellulosic hydroxyl groups, which, in turn, partially prevents these nanoparticles from easily leaching the surface of the cellulosic fabric.

To confirm these assumptions, the surface of the treated cellulosic fabric was elementally analyzed using energy dispersive X-ray (EDX, Figure 7C) to find out what elements it contained. The obtained EDX results (Figure 7C) show that the surface of the treated cellulosic fabric contains the following elements: carbon, oxygen, and zinc. The presence of carbon and oxygen is assigned to the chemical composition of cellulose itself and the stabilizer (Sodium alginate). Meanwhile, the presence of zinc is due to the deposition of ZnONPs on the surface of the fabric. All this confirms that ZnONPs are fixed on the surface of the fabric even after the repeated washing process.

### 3.3. Antimicrobial Efficiency of the Treated Cotton Fabrics

For using the ZnONPs@CF as a wound dressing material, it is necessary to evaluate these materials against microbial species such as *S. aureus*, *E. coli*, *C. albicans*, and *A. niger*. The bactericidal impact of the treated cellulosic materials was investigated using the agar diffusion method. Antimicrobial activity was assessed: the microbial growth below and above the surface cellulosic cotton fabric, as well as the existence of at least 1 mm of inhibition zone surrounding the fabric. The obtained data can be visualized from Figure 8 and Table 1. There is no zone of inhibition for either strain for the untreated cellulosic fibric (CF). Furthermore, microbes were found to thrive beneath the specimens, demonstrating the absence of any antimicrobial activities.

Conversely, all cellulosic fabrics treated with different concentrations of ZnONPs (ZnONPs-1@CF, ZnONPs-2@CF, and ZnONPs-3@CF) exhibit a clear zone of inhibition around the ZnO-coated cellulosic fabrics in contact with *aureus*, *E. coli*, *C. albicans*, and *A. niger*, indicating the complete killing and inhibiting of the microbial growth or replication. Because of the difference in inhibitory zone diameter, the ZnONPs-2@CF appear to have higher antimicrobial effects against *S. aureus*, *E. coli,* and *C. albicans* than against *A. niger*. The variation in cell walls between *S. aureus*, *E. coli, C. albicans,* and *A. niger*, as well as *C. albicans*, makes them more sensitive to ZnONPs [27]. Overall, the results show that the nanoparticles loaded cellulosic fabrics exhibit excellent antibacterial activity against the tested microbes, facilitating their application as wound dressing materials.

### 3.4. Wound Healing of Diabetic Rats Treated with ZnONPs@CF

It is well known that diabetes is recognized to be a disease of disturbed glucose homeostasis, with chronic hyperglycemia resulting in advanced glycation end products, which seem to be principally vital for the cell destruction and also have a sluggish turnover rate. Hyperglycemia causes an increase in the generation of reactive oxygen species and cytochrome C release, which results in caspase-3 activation and cardiac cell death [28]. Myocardial cell death is nearly entirely prevented by partial regulation of elevated glucose levels through insulin; alternatively, it may also be asserted that hyperglycemia causes a large rise in apoptosis. Apoptosis dysregulation as a consequence of hyperglycemia also seems to be widespread, resulting in the impaired healing of infected or diabetic wounds [29,30].

In the case of diabetic patients, the process of the wound healing is time consuming, and it is hampered instead of stopped [31]. A non-healing wound is more dangerous, which is a major reason behind the healing process taking a longer time. Functional constraints can include altered gait and difficulties walking, infections, and, in certain cases, potential malignant alterations [32]. Chronic wounds are susceptible to malignant alterations that are described as a Marjolin’s ulcer, which is a kind of aggressive squamous cell carcinoma [33]. Thus, we aimed to prepare efficient wound dressing materials based on cellulosic fabrics treated with small size ZnONPs. The cellulosic fabrics loaded with different concentrations of ZnONPs were in-vivo evaluated as dressings for the diabetic rats. Firstly, the rats were injected with STZ to become diabetic rats, and then the wounds of these diabetic rats were unable to heal even after a long time. In addition to the lack of healing process, these wounds can become inflamed and infected with time.

After 7 and 21 days of treatment, the percentages of wound contraction for all treated rats are illustrated in Figure 9. As observed, percentages of wound contraction for the diabetic group that was maintained with only cotton dressing without any treatment (blank group) was only 48%, as depicted from Figure 9a,b. The percentage of healing increased upon covering the rat wounds daily with ZnONPs@CF. After 27 days of treatments, the obtained data revealed that the percentage healing for the wounds treated with ZnONPs-1@CF, ZnONPs-2@CF, and ZnONPs-3@CF are 78.9%, 93.5%, and 87.6%, respectively (Table 2). It can be noted that the healing percentage for the diabetic rats treated with ZnONPs-2@CF is greater than that of ZnONPs-1@CF and ZnONPs-3@CF, which could be mainly attributed to the effect of both concentration and particle shape of ZnONPs-2. Thus, ZnONPs-2@CF gave the best effect and showed high wound contraction rates.

As previously mentioned, ZnONPs-2 exhibit spherical shape and small size compared to that of ZnONPs-3@CF. The small particle size leads to the formation of highly efficient particles with greater surface area. These two parameters can facilitate the opportunity of ZnONPs-2@CF rather than ZnONPs-1@CF and ZnONPs@CF. However, the rats treated with ZnONPs-3@CF-3 have significant healing process compared to ZnONPs-1@CF, which is also suitable to be used as wound dressing material.

According to the findings, ZnONPs provide a greater degree of biodegradability and biocompatibility under physiological settings and could be used as a material for wound healing in the treatment of various wounds [34]. ZnONPs are metal-based nanoparticles that have received a lot of interest for their physico-chemical and biological characteristics in wound healing applications. ZnONPs’ unique intrinsic features stimulate wound healing and efficiently restrict the development of microbes at the wound site, and such nanoparticles are suitable for treating acute and chronic wounds [35].

### 3.5. Histopathological Properties of the Treated Wounds

On examination of normal skin (Figure 10a) and non-diabetic, non-treated wounds (Figure 10c) showed that the normal skin consisted of epidermis, dermis, and hypodermis. The stratum corneum appeared with packed horny layers. The dermis appeared with dense matrix. On the other hand, non-treated diabetic wounds (Figure 10b) showed many histological alterations compared to control skin. There, epidermis appeared with disarranged stratum corneum and widely separated individual horny layers. Additionally, the hair follicles and the sebaceous glandes appeared deformed in the disorganized matrix of the dermis.

The above-described histological changes, in non-treated diabetic wounds, were reduced in the diabetic wound treated with ZnONPs-2@CF compared with non-treated diabetic wound. The skin appeared with improved structure where the individual horny layers appeared less separated, the hair follicles and the sebaceous glands appeared more preserved, and the dermal matrix appeared denser and more organized, indicating that treatment with ZnONPs-2@CF has the ability to partially reduce the histological abnormality in the skin structure of diabetic rats.

Under the dermis and hypodermis layers of the skeletal muscle, hyalinization was also observed. Figure 10c clearly displays hyperkeratosis and parakeratosis in the epidermal layer, a flattening thick horny layer, a decrease in sebaceous gland size, and hair follicles with a narrow lumen and pyknotic nucleus in an untreated diabetic rat. On the other hand, Figure 10d depicts a slice of skin from a diabetic rat treated with ZnONPs-2@CF, demonstrating that the epidermis and dermis seemed to be normal in structure. The epidermal and dermal layers were well delineated. The keratin layer was well-formed and located just under the top layer of the epidermis. There were no inflammatory cells in the dermis.

## 4. Conclusions

Three different concentrations of ZnONPs (ZnONPs-1, ZnONPs-2, and ZnONPs-3) were prepared using the solid-state technique. The preparation took a few minutes, and involved eco-grinding artlessly for sodium alginate powder, sodium hydroxide beads, and zinc acetate without the use of solvent. Sodium alginate is considered as an efficient stabilizing agent for the formed ZnONPs-1 and ZnONPs-2, and protects them from remarkable agglomeration. The three formed nanoparticles were used for cellulosic fabric treatments. SEM showed that the surface structure of the cellulosic fabrics was changed to rough form due to the physical deposition of these nanoparticles onto the cellulosic surface. Additionally, ZnONPs-cellulosic fabrics exhibit superior bactericidal features against the tested microbes. Ultimately, the wound healing of ZnONPs@CF displayed efficient healing for the diabetic rat wounds after 21 days of treatment, and the wound contraction percent displayed a value more than 90% with no toxicity on the skin as noted by the histopathological studies, indicating that ZnONPs@CF-2 has the ability to partially reduce the histological abnormality in the skin structure of diabetic rats. Therefore, ZnONPs@CF can be widely used in dressings for diabetic wounds, as the method of preparation and application is facile and environmentally safe and does not cause any problems that impede its use.

## Figures and Tables

**Figure 1 polymers-14-02168-f001:**
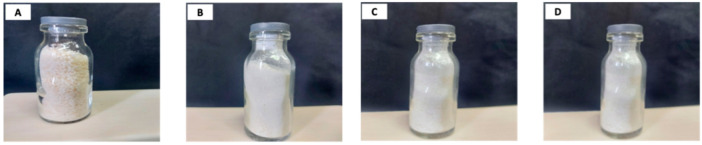
Photo images of (**A**) sodium alginate, (**B**) ZnONPs-1, (**C**) ZnONPs-2, and (**D**) ZnONPs-3 synthesized in their solid form.

**Figure 2 polymers-14-02168-f002:**
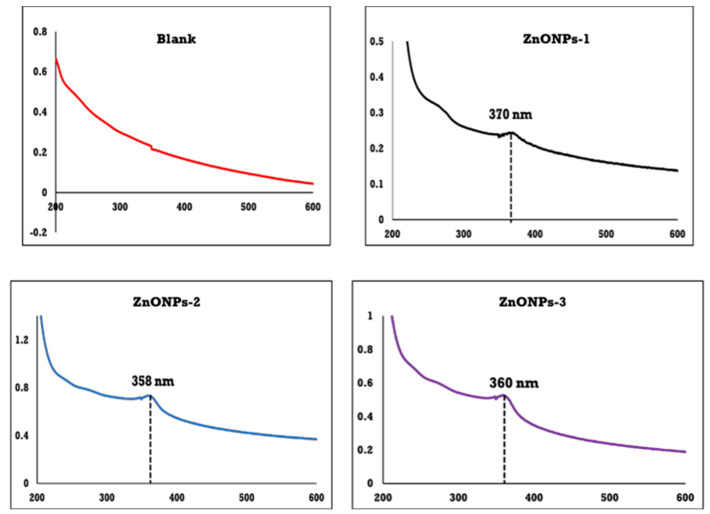
UV-vis of sodium alginate (blanks) and ZnONPs (ZnONPs-1, ZnONPs-2, ZnONPs-3).

**Figure 3 polymers-14-02168-f003:**
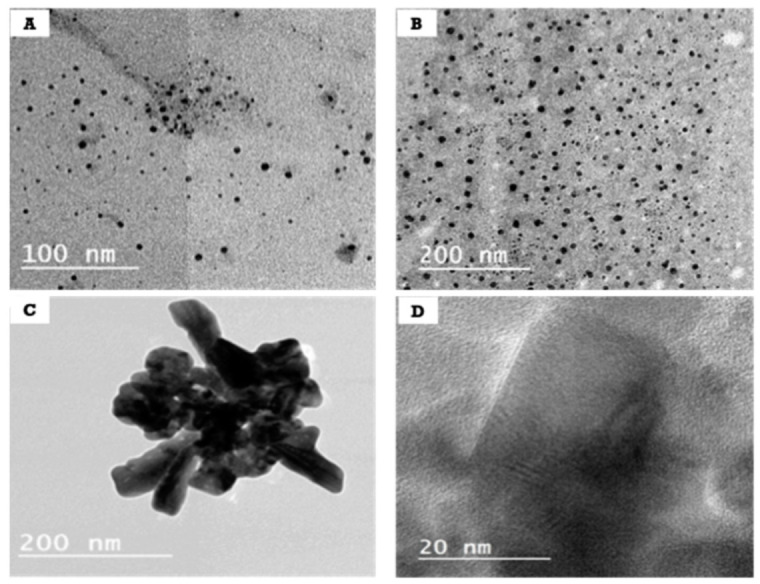
TEM images of (**A**) ZnONPs-1, (**B**) ZnONPs-2 and (**C**) ZnONPs-3, and (**D**) high-resolution TEM image of ZnONPs-2.

**Figure 4 polymers-14-02168-f004:**
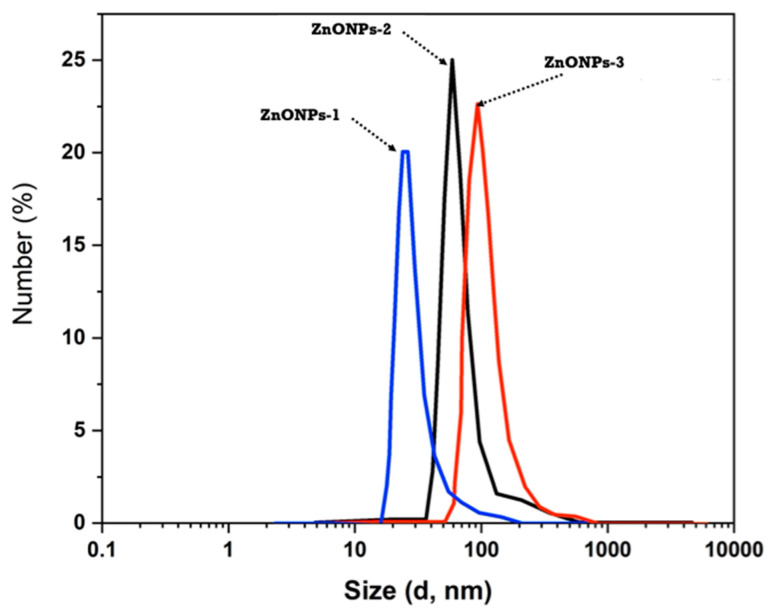
Particle size analysis of ZnONPs-1, ZnONPs-2 and ZnONPs-3.

**Figure 5 polymers-14-02168-f005:**
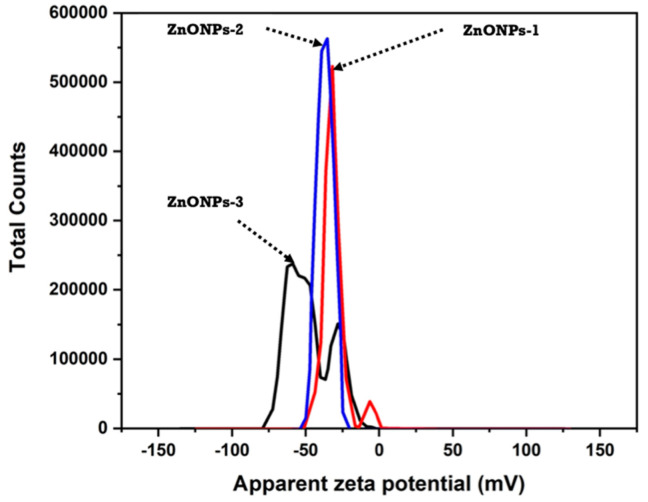
Zeta potential value of ZnONPs-1, ZnONPs-2 and ZnONPs-3.

**Figure 6 polymers-14-02168-f006:**
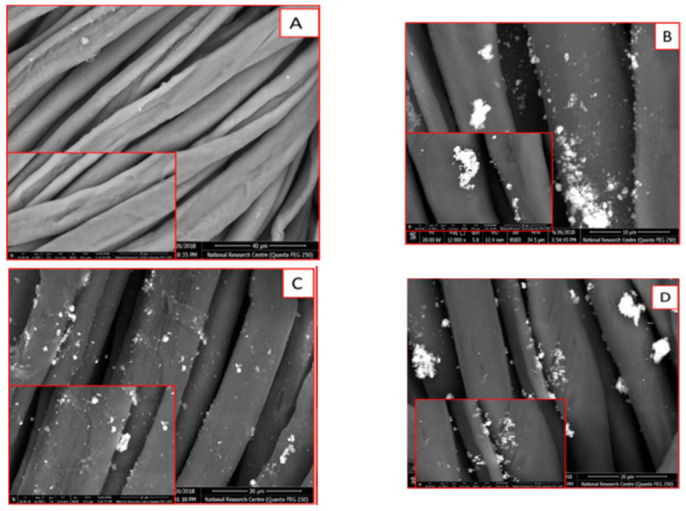
SEM images of (**A**) untreated fabric, (**B**) ZnONPs-1@CF, (**C**) ZnONPs-2@CF, and (**D**) ZnONPs-3@CF.

**Figure 7 polymers-14-02168-f007:**
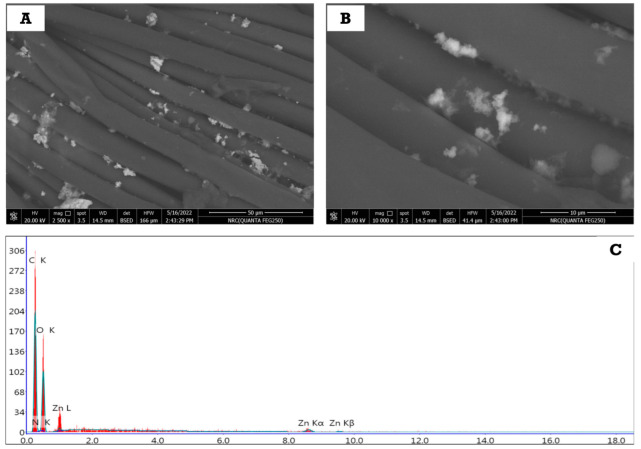
SEM at (**A**) 2500× and (**B**) 10,000×); (**C**) EDX of the treated cellulosic fabric with ZnONPs-2.

**Figure 8 polymers-14-02168-f008:**
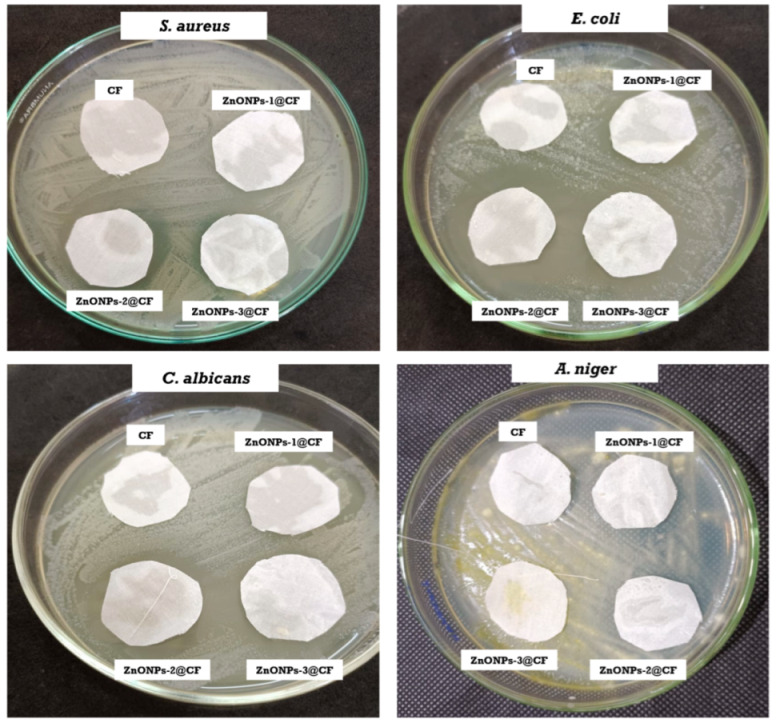
Inhibition zone evaluation of blank and ZnONPs@CF against pathogenic microbes.

**Figure 9 polymers-14-02168-f009:**
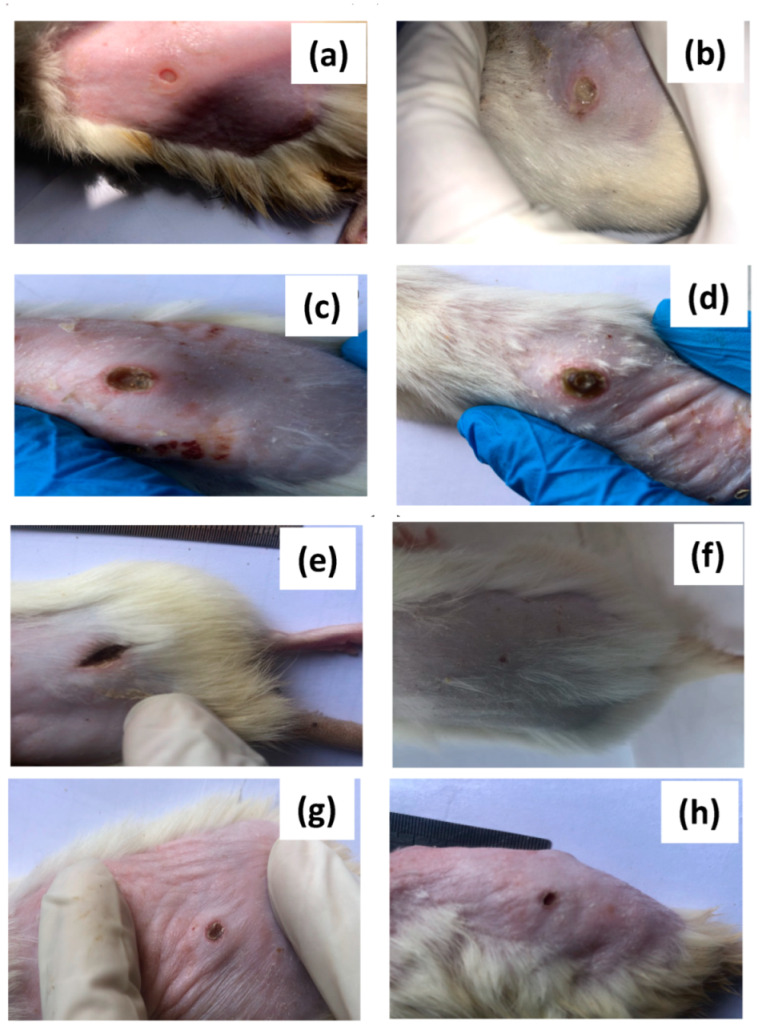
Wounded skin contraction in experimental rats after 7 days and the end of experiment (21 days). (1 and 2) revers to the treatment after 7 and 12 days, respectively. When (**a**,**b**) refers to the rats treated with cotton fabric (blank), (**c**,**d**) refers to the diabetic rats treated with ZnONPs-1@CF, (**e**,**f**) refers to the diabetic rats treated with ZnONPs-2@CF, and (**g**,**h**) refers to the diabetic rats treated with ZnONPs-3@CF.

**Figure 10 polymers-14-02168-f010:**
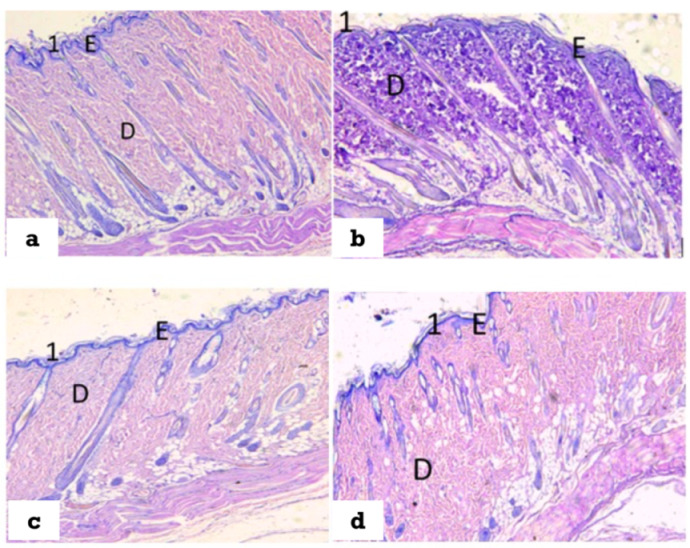
Section of skin: (**a**) normal skin, (**b**) diabetic wound, (**c**) non- diabetic non -treated wound, and (**d**) diabetic wound treated with ZnONPs-2@CF; H&E stain, Scale Bar: 500 µ. 1: stratum corneum, E: epidermis, D: Dermis.

**Table 1 polymers-14-02168-t001:** The average diameter zone (mm) for the untreated cellulosic fabric and ZnONPs@CF against the tested pathogenic microbes.

Sample Code	Inhibition Clear Zone Diameter (mm)			
	*S. aureus*	*E. coli*	*C. albicans*	*A. niger*
CF	0	0	0	0
ZnONPs-1@CF	14	18	13	14
ZnONPs-2@CF	18	23	19	21
ZnONPs-3@CF	14	16	17	18

**Table 2 polymers-14-02168-t002:** Wound healing percent of diabetic rats treated with ZnONPs@CF.

Groups	Wound Contraction (%)
Group I (CF) (a, b)	27%
Group II (ZnONPs-1@CF) (c, d)	78.9%
Group III (ZnONPs-2@CF) (e, f)	93.5%
Group IV (ZnONPs-3@CF) (g, h)	87.6%

## Data Availability

All relevant data are within the manuscript and available from the corresponding author upon request.

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
