# Peer review of "Antidiabetic Wound Dressing Materials Based on Cellulosic Fabrics Loaded with Zinc Oxide Nanoparticles Synthesized by Solid-State Method"

_polymers, 2022, doi:10.3390/polym14112168_

Round 1

Reviewer 1 Report

This manuscript presented quite interesting results but the quality of presentation must be improved:

1) Antimicrobial tests.i cannot see the inhibition zones on your prictures. Please provide better pictures with a bar providing the size of inhibition zones

2) Characterization. Your ZnO nanoparticles are not sufficiently charscterized, especially their linkage to the fabrics. Please investigate their stabilityon the fabrics in water and PBS. E g. Provide SEM after washing.

3) Histology. The results are not sufficiently described.  Please improve it.

4) There are many typos, e.g you wrote Zno with small O. Check the text. Firmatting is also uncorrect.

Author Response

Reviewer #1

This manuscript presented quite interesting results but the quality of presentation must be improved:

1) Antimicrobial tests. i cannot see the inhibition zones on your prictures. Please provide better pictures with a bar providing the size of inhibition zones.

Reply:

Thanks for your comment. Antimicrobial tests were repeated to have clear pictures for the inhibition zone as Shown in Figure 8, please see the revised manuscript.

2) Characterization. Your ZnO nanoparticles are not sufficiently charscterized, especially their linkage to the fabrics. Please investigate their stability on the fabrics in water and PBS. E g. Provide SEM after washing.

Reply:

Thanks for your comment. The aim of our research work is to prepare ZnONPs using facile and efficient method without noticeable financial cost, which can be used in treatment of the cellulosic fabrics to be acted as dressings for the diabetic wounds. The as-prepared ZnONPs have been completely characterized by Uv-vis spectroscopy, TEM, Particle size analyzer and Zeta potential. Meanwhile, cellulosic fabrics loaded with ZnONPs were characterized by SEM technique. The treated cellulosic fabrics acted as a carrier for ZnONPs, which is physically adsorbed onto the surface of cellulosic fabrics. Upon using, these nanoparticles can be easily leached out from the cellulosic surface to treat the diabetic wounds and thus killing the pathogenic microbes, which in turn, facilitate the healing of diabetic wounds. These bandages can be used once only. It is worth noting that during the treatment of diabetic wound, which took place for 27 days on experimental rats, the bandages were changed daily and replaced with a new one that is placed on the diabetic wound. So, there is no need to wash the bandages (disposable bandages).

However, the as-prepared cellulosic fabrics loaded with ZnONPs-2 was selected to evaluate the durability of the fabric after many washing cycles. Firstly, the treated fabric was washed with distilled water using traditional detergent for 75 min in washing machine at 40 °C. Then, the washed fabric was squeezed and dried in air followed by assessing the morphological features using FESEM. Figure 7 (A, B) illustrates the topographical surface of the treated cellulosic fabrics after washing. The scanned for the treated fabrics was evaluated at different magnifications; 2500 X (Figure 7 A) and 10000 X (Figure 7 B). It can be observed that from the two images that the nanoparticles are still deposited onto the surface of the fabrics which can be ascribed to the good adhesion of nanoparticles to the surface of cellulosic fabrics. As mentioned, ZnONPs was prepared using sodium alginate as stabilizing agent to cover and coat these particles and prevent them from agglomeration, thus, the covered layer (sodium alginate) is able to chemically bonded with cellulosic hydroxyl groups, which, in turn, partially prevents these nanoparticles from easily leaching the surface of the cellulosic fabric.

3) Histology. The results are not sufficiently described.  Please improve it.

Reply:

Thank you for your comment. The results of Histology part were revised and improved. Please see the revised manuscript.

4) There are many typos, e.g you wrote Zno with small O. Check the text. Firmatting is also uncorrect.

Reply:

Thanks for your notes. The whole manuscript was revised carefully, all abbreviations have been checked and amended as required by your suggestion.

Reviewer 2 Report

The manuscript entitled “Antidiabetic wound dressing materials based on cellulosic fabrics loaded with zinc oxide nanoparticles synthesized by solid-state method” reported the synthesis of ZnO nanoparticles by a facile solid-state reaction, which were further loaded onto cellulosic fabrics as wound dressing. The amount of zinc precursor was varied to produce different ZnO nanoparticles. The as-synthesized ZnO nanoparticles were characterized by UV-vis spectroscopy, TEM, DLS, and zeta potential analysis. In addition, antimicrobial efficiency and wound healing ability of ZnO/cotton fabrics were assessed. The loading of ZnO on cotton fabrics resulted in the wound dressing with an efficient healing for diabetic rat wounds. The result overall is interesting, and ZnO/cotton fabrics is promising for wound healing. However, the manuscript still needs great improvement. The following points should be addressed satisfactorily before the manuscript could be further considered for publication.

  1. It would be important to review recent progress in wound dressings using ZnO nanoparticles.
  2. The selected area diffraction could not support that ZnO is single crystalline because there appear rings rather than spots. A high-resolution TEM for a representative ZnO nanoparticle would be more convincing.
  3. The caption of Figure 8 is unclear. What do a)-d) refer to?
  4. Figure 8a1-a2: It does not seem that the same rat was used for the evaluation of wound healing. The authors need to clarify it.
  5. Figure 8c2: A zoomed-in image would be better to show the wounded skin.
  6. There appear two c2 and two d2 in Figure 8.
  7. Figure 9: what’s the difference between diabetic wound (Figure 9b) and untreated diabetic wound (Figure 9c)?
  8. Figure 9: the description of normal skin (Figure 9a) and untreated diabetic wound (Figure 9c) appear to be the same, which is obviously incorrect.
  9. What’s ratio of ZnO nanoparticles in the wound dressing? Does the ratio of ZnO have any effect on the antimicrobial and wound healing ability of wound dressing?
  10. Typos need to be corrected: Materials section, “homocystien” to “homocysteine”; “mo0dification” to “modification”; Discussion section 3.1, add “spectroscopy” or “spectrum/spectra” after “UV-vis”; “ZnO ions” to “ZnO”; “zin salt” to “zinc salt”; “the fabrication of highly throughput ZnONPs” to “the high-throughput fabrication of ZnONPs”; Section 3.2, “facilate” to “facilitate”; add “image” after “Figure 6 shows SEM”.
  11. A full name of “STZ” is needed.

Author Response

Reviewer #2

Comments and Suggestions for Authors

The manuscript entitled “Antidiabetic wound dressing materials based on cellulosic fabrics loaded with zinc oxide nanoparticles synthesized by solid-state method” reported the synthesis of ZnO nanoparticles by a facile solid-state reaction, which were further loaded onto cellulosic fabrics as wound dressing. The amount of zinc precursor was varied to produce different ZnO nanoparticles. The as-synthesized ZnO nanoparticles were characterized by UV-vis spectroscopy, TEM, DLS, and zeta potential analysis. In addition, antimicrobial efficiency and wound healing ability of ZnO/cotton fabrics were assessed. The loading of ZnO on cotton fabrics resulted in the wound dressing with an efficient healing for diabetic rat wounds. The result overall is interesting, and ZnO/cotton fabrics is promising for wound healing. However, the manuscript still needs great improvement. The following points should be addressed satisfactorily before the manuscript could be further considered for publication.

  1. It would be important to review recent progress in wound dressings using ZnO nanoparticles.

Reply:

Thanks for your comment. A complete survey has been done and mentioned in Introduction part to cover the wound dressing based ZnONPs.

Regarding the in situ fabrication, Sun et al., (Sun et al. 2016) produced bactericidal ZnONPs loaded cotton fabric using ultrasonic irradiation in which ZnONPs was synthesized with the aid of amino-terminated silicon sol. Such study described a technique for the deposition of nanostructured ZnONPs onto cotton fabric to introduce antibacterial advanced features. Their findings revealed that ZnONPs loaded cotton fabric had extremely good UV protection and remarkable antibacterial characteristics. Recently, Sun X et al., (Sun and Zhang 2021) established an in-situ synthesis technique for ZnONPs to be used for the producing of bactericidal fabrics for healing domains. Their findings revealed that the fabric containing ZnONPs had absolutely superb UV protection, efficient antimicrobial functions, as well as modest cytotoxicity. Angélica et al., (Lund et al. 2018) create bactericidal cotton gauze fabrics, which are chemically modified through graft copolymerization with poly(methacrylic acid). After that, ZnONPs were added to the grafted cotton gauzes to give them bactericidal features.

  1. The selected area diffraction could not support that ZnO is single crystalline because there appear rings rather than spots. A high-resolution TEM for a representative ZnO nanoparticle would be more convincing.

Reply:

High-resolution TEM for ZnONPs-2 has been added and discussed in the revised manuscript.

  1. The caption of Figure 8 is unclear. What do a)-d) refer to?

Reply:

The caption of Figure 8 has been amended and illustrated under the figure.

  1. Figure 8a1-a2: It does not seem that the same rat was used for the evaluation of wound healing. The authors need to clarify it.

Reply:

Thanks for your comment. However, the picture has been taken for the rat after 27 days of treatment. This group was treated with the untreated cotton fabrics. The untreated cotton fabric has no positive impact on the wound. However, it does some inflammations for the wound and change the color of the wound and skin due to the inflammation properties.

Figure 8c2: A zoomed-in image would be better to show the wounded skin.

Reply:

The image has been replaced. Taking into account that the shooting angle for each mouse is different and also varies with the passage of days. In addition, skin color and hair volume vary over the course of treatment days

  1. There appear two c2 and two d2 in Figure 8.

Reply:

Thanks for your observation and we are sorry for this mistake. The caption and ligand of all images have been amended.

  1. Figure 9: what’s the difference between diabetic wound (Figure 9b) and untreated diabetic wound (Figure 9c)?

Reply:

Thanks for your observation. It is typing mistake and corrected description was added. Figure 9b is the non-treated diabetic wound and Figure 9c is treated non-diabetic wound

  1. Figure 9: the description of normal skin (Figure 9a) and untreated diabetic wound (Figure 9c) appear to be the same, which is obviously incorrect.

Reply:

Yes, as I mentioned above normal skin (Figure 9a) while Figure 9c is treated non-diabetic wound. Structurally in other sections with similar magnification both are similar in structure of epidermis and dermis compared to the altered figure 9b (non-treated diabetic, We corrected this mistake by using the same magnification for all used micrographs.

  1. What’s ratio of ZnO nanoparticles in the wound dressing? Does the ratio of ZnO have any effect on the antimicrobial and wound healing ability of wound dressing?

Reply:

Thanks for your comment. As mentioned in our manuscript, we prepared three different concentrations of ZnONPs using different concentrations of the zinc precursor (1.5, 3 and 4.5 g). Then the prepared nanoparticles were nominated as ZnONPs-1, ZnONPs-2 and ZnONPs-3 attributed to the utilized concentrations of zin acetate. After the preparation of ZnONPs using the more efficient solid-state technique, 1g of each concentration (ZnONPs-1, ZnONPs-2 and ZnONPs-3) was dissolved in deionized water (100 mL) under the effect of sonication for 15 min. The bleached cotton fabric (10 cm × 10 cm) was dipped in each solution for 1 min and squeezed with pick up 80%. The treated cotton fabrics were dried at 90 oC for 7 min and curing at 150 oC for 3 min.

Thus, each sample of the treated cellulosic fabrics with ZnONPs-1, ZnONPs-2 and ZnONPs-3 contain different concentration of ZnONPs. As depicted from our findings, the various results obtained as antimicrobial fabrics are definitely due to the concentration of ZnONPs deposited on the surface of the fabric. The higher the concentration of ZnONPs on the surface of the fabric, the more resistant it is to the infectious microbes.

Taking into account the size of the formed ZnONPs and its ability to be stable against the agglomeration into large-sized particles. The aggregated ZnONPs (ZnONPs-3) with large size have lower antimicrobial resistance when compared to smaller size nanoparticles and more stable against aggregation (ZnONPs-2).

This, of course, affects the ability of the treated cellulosic fabrics when used as a dressing for diabetic wounds (Table 2: Wound healing percent of diabetic rats treated with ZnONPs treated cellulosic fabrics.).

  1. Typos need to be corrected: Materials section, “homocystien” to “homocysteine”; “mo0dification” to “modification”; Discussion section 3.1, add “spectroscopy” or “spectrum/spectra” after “UV-vis”; “ZnO ions” to “ZnO”; “zin salt” to “zinc salt”; “the fabrication of highly throughput ZnONPs” to “the high-throughput fabrication of ZnONPs”; Section 3.2, “facilate” to “facilitate”; add “image” after “Figure 6 shows SEM”.

Reply:

Thanks for all these useful observations that enhance our manuscript. All these typos have been amended.

  1. A full name of “STZ” is needed.

Reply:

Thank you for your question, Streptozotocin (STZ) is added to methodology section.

Reviewer 3 Report

The work done by  Elsawy et al is novel and innovative, i will accept this manuscript for publication after minor revision.

  1. The graphical abstract is too lengthy kindly make it short and to the point, Also add the main finding in the abstract
  2. Authors need to draw a graphical abstract to draw the reader's attention.
  3. In the materials section, kindly make separate heading for animal study and clearly describe all animal protocols
  4.  The animal study is approved from ethical Committee, kindly add the date and number for this approval.
  5. Before starting treatment on animals, how the animal wound model was developed author never discussed it, kindly add how this model developed.
  6. what does mean by solid-state in ZnO NPs?
  7. Figure 1 is very poor, I didn't find any difference, kindly upload a high-resolution figure to indicate clearly differences. 
  8. Discussion: This part requires a thorough development. The authors should clarify the signalized doubts. They should demonstrate the advantages and disadvantages of the proposed system against the background of similar systems described earlier. The authors should also present their suggestions related to possible possibilities of the practical application of the described solution.
  9. Please revisit the entire manuscript for minor grammar issues. The writing although good needs to be corrected for grammar and sentence construction. I also highly recommend the authors streamline their writing to keep the underlying conclusions precise and clear. The transitions between ideas seem disconnected. These would only help the reader get more from the review and improve its quality and appeal

Author Response

Reviewer #3

Comments and Suggestions for Authors

The work done by  Elsawy et al is novel and innovative, i will accept this manuscript for publication after minor revision.

  1. The abstract is too lengthy kindly make it short and to the point, Also add the main finding in the abstract

Reply:

Thanks for your comment. The abstract has been rewritten to have more quantitative data as follow:

“The current study aimed for the preparation of high throughput zinc oxide nanoparticles (ZnONPs) via the utilization of solid-state technique which is considered an efficient way for the preparation of ZnONPs with high concentration. Sodium alginate was used as stabilizing agent for the formed ZnONPs with three different concentrations; ZnONPs-1, ZnONPs-2 and ZnONPs-3 (attributed to the utilized different concentrations of zinc acetate (1.5, 3 and 4.5 g)). The resultant depicted that, above the utilization of 3 g of zinc acetate, sodium alginate was not able to stabilize ZnONPs and leads to the formation of agglomerated particles with no clear edges as proved from TEM for ZnONPs-3. In addition, DLS findings illustrated that the as-prepared ZnONPs exhibited average particle size equal to 78, 117 and 144 nm, respectively. The data also depicted that, all the formulated ZnONPs were formed with good stability (above -30 mv). The as-fabricated ZnONPs (ZnONPs-1, ZnONPs-2 and ZnONPs-3) were used for the treatment of cellulosic fabrics and the topographical images (obtained via SEM) affirmed the deposition of nanoparticles onto the surface of cellulosic fabrics with no noticeable agglomeration. The findings also outlined that the treated cellulosic fabrics dressings were proven to have increased bactericidal characteristics against the pathogenic microorganisms. The finding of Wound contraction for the diabetic rats was measured after 21 days and reached 93.5 % after treating the diabetic wound with cotton fabrics containing ZnONPs-2. Ultimately, the generated wound dressing (ZnONPs loaded cellulosic fabrics; ZnONPs@CF) offer considerable promise for treating the wound infections and might be examined as a viable alternative to antibiotics and topical wound treatments.”

  1. Authors need to draw a graphical abstract to draw the reader's attention.

Reply:

Graphical abstract has been drawn and uploaded with the manuscript

  1. In the materials section, kindly make separate heading for animal study and clearly describe all animal protocols

Reply:

Thanks for your comment. It was done in the manuscript.

  1. The animal study is approved from ethical Committee, kindly add the date and number for this approval.

Reply:

Thanks for your comment. It was added to the manuscript.

  1. Before starting treatment on animals, how the animal wound model was developed author never discussed it, kindly add how this model developed.

Reply:

Thanks for your comment. the development of animal wound model has been added as follow:

“Streptozotocin (Stz) was dispersed in sodium citrate solution (50 mM; pH 4.5) comprising NaCl (150 mM) and injected subcutaneously to rats (3 mg/50 g of body weight). Fasting blood sugar was measured after three days for affirming the establishment of diabetes mellitus. If the animals' fasting glucose level was higher than 200 mg/dl, they became catalogued as diabetic animal. Diethyl ether (45 %) was used to anaesthetized the diabetic rats. Ethanol (70%) was used to sterilize the wounded area (shaved area). Taking in mind that, the single longitudinal skin incision has been performed just on shaved area.

The wounded skin rats were evenly dressed with an experimental sterile dressing consisting of cellulosic fabric that loaded with zinc oxide nanoparticles in various concentrations; ZnONPs-1@CF (Group II), ZnONPs-2@CF (Group III) and ZnONPs-3@CF (Group IV) per kg b.w. Taking into mind that, the negative control group was kept with untreated cotton materials that were not treated in any way (Group I). This experiment took place over the course of 21 days. Daily, the animals were treated with a new sample of the treated cotton fabrics that was appropriate for each group.”

  1. What does mean by solid-state in ZnO NPs?

Reply:

Thanks for your comment. As known, there are many techniques used for the preparation of nanoparticles. in our manuscript, we used on the promising techniques for such preparation. Solid state synthesis means that the preparation was carried out in the absence of aqueous or organic solvents. We just grinding all precursors (Zinc acetate, sodium hydroxide and sodium alginate) with each other’s.

The advantages of this method are that we can prepare the nanoparticles at very high concentrations in a short time and without consuming energy and at a very cheap economic cost. In addition, by using this method, the as-prepared nanoparticles can be easily transferred to industrial companies or factories that use them in the fabric treatment, unlike nanoparticles in their liquid form, which require large containers to be packaged and transported, which increases the cost of production. As well as the use of liquids and solvents in the preparation also increases the cost of preparation.

  1. Figure 1 is very poor, I didn't find any difference, kindly upload a high-resolution figure to indicate clearly differences. 

Reply:

Thanks for your comment. There is no difference between all images. We just want to show that by using this method of preparation, we get nanomaterials with a high concentration that can be used in many fields. This quantity can be used to prepare many meters of cotton fabrics without increasing the burden or effort. However, we imaged these nanometric materials from a different angle for illustration only.

  1. Discussion: This part requires a thorough development. The authors should clarify the signalized doubts. They should demonstrate the advantages and disadvantages of the proposed system against the background of similar systems described earlier. The authors should also present their suggestions related to possible possibilities of the practical application of the described solution.

Reply:

The advantages of solid-state synthesis are that we can prepare the nanoparticles at very high concentrations in a short time and without consuming energy and at a very cheap economic cost. In addition, by using this method, the as-prepared nanoparticles can be easily transferred to industrial companies or factories that use them in the fabric treatment, unlike nanoparticles in their liquid form, which require large containers to be packaged and transported, which increases the cost of production. As well as the use of liquids and solvents in the preparation also increases the cost of preparation.

  1. Please revisit the entire manuscript for minor grammar issues. The writing although good needs to be corrected for grammar and sentence construction. I also highly recommend the authors streamline their writing to keep the underlying conclusions precise and clear. The transitions between ideas seem disconnected. These would only help the reader get more from the review and improve its quality and appeal

Reply:

Thanks for your comment and we are sorry for these mistakes. The whole manuscript has been revised and typos amended.

Round 2

Reviewer 1 Report

Authors have made sufficient improvements including the revised Figures and expanded discussion. The paper can be now accepted.

Reviewer 2 Report

The authors have addressed my comments. The manuscript can be accepted in present form.